# The use of ultraviolet light generated from light-emitting diodes for the disinfection of transvaginal ultrasound probes

Muhammad Yasir👤*, Mark D. P. Willcox👤

School of Optometry and Vision Science, University of New South Wales, Kensington, Australia

* m.yasir@unsw.edu.au, yasirjri85@gmail.com

## Abstract

Transvaginal ultrasound probes (TVUS) are used for several gynecological procedures. These need to be disinfected between patient use. In the current study we examine whether UVC delivered using light emitting diodes for 90 seconds can provide sufficient disinfection efficacy. A new UVC device that delivers UVC radiation at 265nm-275nm for 90 seconds was used. TVUS probes were swabbed before and after use in an *in vitro* fertilization clinic. Microbes on the swabs were cultured and identified. In addition, the ability of the UVC device to provided repeated high-level disinfection was analysed by deliberately contaminating probes with spores of *Bacillus subtilis* and then performing the UVC disinfection and bacterial culture. 50% of probes were contaminated with bacteria, most commonly *Bacillus* sp., directly after *in vivo* use. Whereas 97% were sterile after UVC disinfection for 90 seconds. The UVC treatment resulted in no growth of *B. subtilis* spores after each of five repeated contaminations with 5–9 x $10^7$ spores on the probes. This study has found that UVC delivered via light emitting diodes for only 90 seconds can produce high level disinfection of transvaginal probes.

## Introduction

The use of transvaginal ultrasound (TVUS) has become an indispensable tool in contemporary gynecological practice, enabling the diagnosis and monitoring of various reproductive health conditions with enhanced precision. By employing high-frequency sound waves, TVUS provides detailed imaging of the pelvic organs, aiding in the detection of abnormalities, guiding interventions, and facilitating better patient outcomes. However, the increasing use of TVUS also raises concerns regarding the potential transmission of infections, emphasizing the critical need for thorough disinfection practices to safeguard women's health.

The intimate nature of TVUS, involving direct contact between the ultrasound probe and the vaginal mucosa, poses a unique challenge in terms of infection prevention. The vaginal canal harbors a diverse microbial ecosystem, including bacteria, viruses, and fungi, both beneficial and potentially pathogenic. Generally, the vaginal microbiota includes *Gardnerella vaginalis*, *Ureaplasma urealyticum*, *Candida albicans*, *Prevotella* spp., and *Lactobacillus* spp [1, 2]. If high level disinfection protocols are not adhered to, the transfer of microorganisms between

**Data Availability Statement:** All data are provided with the submitted manuscript.

**Funding:** This research was funded by Lumicare, Sydney, Australia.

**Competing interests:** The authors have declared that no competing interests exist.

patients via contaminated probes can occur, potentially leading to nosocomial infections [3, 4]. A meta-analysis of contamination rates on endovaginal/rectal probes found a prevalence rate of 12.9% for potentially pathogenic bacteria and 1.0% for viruses after probes were subjected to low-level disinfection [3].

Transvaginal ultrasound probes are typically disinfected using a standardized process that involves several steps to ensure effective decontamination. The specific disinfection procedures may vary between healthcare facilities, but the general principles remain consistent. Pre-use, the probes should be covered with a sterile, single-use cover or a condom [5]. After each use, the cover is removed and the probe is first pre-cleaned to remove any visible organic debris or bodily fluids [5, 6]. This step is usually performed using disposable wipes or dampened cloth with a mild detergent or enzymatic cleaner. Following pre-cleaning, the probe undergoes disinfection to eliminate any remaining microorganisms [5, 6]. Low-level disinfection, using disinfectant-impregnated wipes alone does not eliminate microbial contamination, [7–10], and so high-level disinfection methods are used that include chemical disinfection with, for example, hypochlorite, or peracetic acid, or hydrogen peroxide-based solutions, or ultraviolet-C (UVC) light [6]. Glutaraldehyde, orthophthalaldehyde, phenols, and isopropyl alcohol have been shown to have virtually no efficacy against human papillomavirus [5], but UVC is effective against this virus [11]. The high-level disinfection systems can be automated with such systems employing UVC or hydrogen peroxide vapor.

To achieve such a high-level degree of disinfection, there exist currently four processes capable of fulfilling this requirement. These processes are chemical soaking, chemical aerosol, surface wiping, and UV-C irradiation. Chemical soaking is a process that requires placing the ultrasound transducer such that it is immersed into a chemical reagent. Such processes generally require a soaking time for the transducer to be left immersed in the chemical reagent of between 8 minutes to 45 minutes. Whilst the appropriate level of disinfection may be achievable, the disadvantage of this process is that the chemical reagent is hazardous and any exposure to the chemical reagent may harm the operator and patient and disposal of the chemical waste may harm the environment. Further, as care is required in handling the chemicals, this method is manually operated and time-consuming [12].

Chemical aerosol is a process whereby the ultrasound transducer is placed within a chamber that is flooded with nebulized hydrogen peroxide. Typically, the transducer is placed within the chamber for between 7 to 12 minutes, depending on the specific conditions [13]. An automated system that uses hydrogen peroxide vapour resulted in 91.4% of probes having no microbial growth compared to 78.8% with manual disinfection [14]. Once again, due to the use of the chemical reagent, the disadvantage of this method is that the residual of chemical reagent left on transducers may harm the operators and patients.

It is possible to achieve the desired level of disinfection through the use of surface wipes. Such a process uses different chemical wipe combinations to manually wipe the surface of the transducer. The procedure requires steps of pre-cleaning, disinfection and rinsing [15]. However, a drawback with such a method is that it requires manual application and is prone to human error, is costly and is time intensive.

The use of ultraviolet-C light (UVC), of wavelengths 255-285nm, has the advantage that it does not use potentially hazardous chemicals nor leave residual chemicals. Also, UVC disinfection has been shown to reduce the time needed to disinfect transvaginal probes [16] and to reduce bacteria and viruses on probes to below detectable levels [17]. Current UVC and chemical disinfectant systems require 5 to 7 minutes of disinfection [14, 17]. Moreover, chemical disinfectant systems are expensive and difficult to use [11].

The aim of the current study was to examine the ability of a disinfection system that uses LED lights to deliver UVC radiation at 265nm-275nm, and that has been designed to require

only 90 seconds to deliver sufficient UVC for high level disinfection, to reduce the numbers of microbes on probes *ex vivo* and in laboratory studies. Our hypothesis was that LED lights can deliver UVC fast and effectively to reduce the viability of microbes on transvaginal ultrasound probes.

## Materials and methods

### *Ex vivo* testing

This study was approved by the Human Research Ethics Committee of the University of New South Wales (HREC number: HC220136). The ultrasound probes were collected after use in an *in vitro* fertilization clinic (Connect IVF, Sydney, Australia) from June 10 to June 27, 2022. Patients consent was obtained on consent forms that included their names and signatures. The ultrasound probes had been sheathed with non-sterile latex free probe covers (Sonologic, Brisbane, QLD, Australia) and coated with gel prior to use. After use, the sheath was removed, and sterile cotton swabs (Bacto Laboratories, Australia) were passed twice over a 5–10 cm$^2$ area from one end to the other of the ultrasound probes on one side (randomly selected) to examine the numbers and types of microbes present on the endoscopes prior to disinfection. We chose not to perform a cleaning step after use of the probes as they had been sheathed in a protective cover which was removed, and whilst cleaning is recommended prior to disinfection [6], in certain instances >10% of operators report either having no policy on cleaning or the policy being impractical [9]. Therefore, the procedure used in the current investigation can be considered as a worst case scenario.

The ultrasound probes were then loaded into the Lumicare ONE UVC LED high level disinfection system (Lumicare, Sydney, Australia; Fig 1) which delivers UVC of 265-275nm for a standard time of 90 seconds. The machine was developed to deliver sufficient UVC for high level disinfection over 90 seconds to reduce the time of disinfection from the usual ≥5 minutes disinfection with other systems [14, 17]. Furthermore, a previous UVC disinfection system used a similar number of seconds to disinfect transvaginal probes and this rapid disinfection was preferred by the majority of healthcare professionals [16]. After disinfection, the ultrasound probe was removed, and the opposite side swabbed, as described above. The swabs were then put into AMIES transport media (Bacto Laboratories, Australia) and transported to the microbiology laboratory of the School of Optometry and Vision Science, University of New South Wales (UNSW) in cold chain and processed within in 4 hours of collection.

To assess whether the latex free probe sheaths had any bacterial contamination, two sheaths were randomly selected and analysed for the presence of bacteria. The sheaths were swabbed from the internal surfaces over a 5–10 cm$^2$ area which would have been directly in contact with the probes.

All the swabs were processed under aseptic conditions. Microorganisms were recovered in autoclaved sterile phosphate buffered saline (PBS; 8 g NaCl, Ajax Finchem, AUS; 0.2 g KCl, Ajax Finchem, AUS; 1.4 g Na$_2$HPO$_4$, Chem Supply, AUS; 0.24 g KH$_2$PO$_4$, Chem Supply, AUS; purity of each ≥ 99%, in 1000 mL of milli Q water) by vortexing the swabs in vials in the presence of glass beads (0.5 mm diameter; Sigma Aldrich St Louis, MO, USA) for 2 minutes. Aliquots (200 μL) of the resulting slurry were spread onto chocolate blood agar plates and incubated in three different atmospheric conditions (aerobic, microaerophilic and anaerobic) for 24–72 hours at 37˚C. An aliquot (100 μL) was also plated onto Sabouraud's agar and incubated for 7 days at 25˚C to grow any fungi present on the endoscopes. Following incubation, numbers of similar colonies of each bacterium and fungus were counted and purified. Pure cultures of microbes were stored in 25% glycerol at -80˚C.

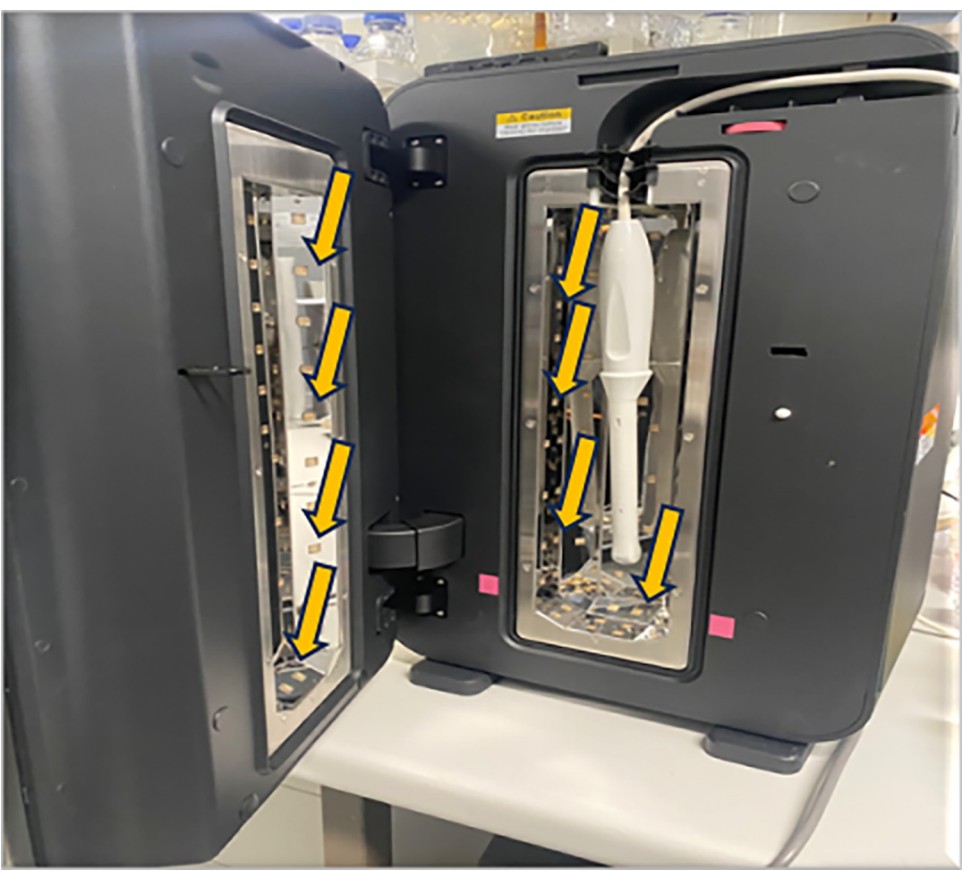

**Fig 1. The Lumicare ONE high level disinfection UVC system that uses LED lights to deliver ultraviolet light C.** A probe has been inserted into the system. The arrows highlight examples of where the LED lights were placed. LED lights were situated on the three sides, top and bottom of the main compartment as well as inside, top and at the bottom of the door. The system delivers UVC of between 265-275nm for 90 seconds and maintains a low temperature (20–25˚C) during disinfection. More information is available here https://lumicare.one.

At a later stage, the microbes were recovered from the -80˚C storage on chocolate blood agar plates by incubation overnight in relevant atmospheric conditions. Following incubation, a loopful of bacterial growth was diluted in 300 μL of high performance liquid chromatography-grade water into an Eppendorf tube and mixed thoroughly until the bacteria were completely suspended in water. This was followed by addition of 900 μL of absolute ethanol with thorough mixing. Then, tubes were centrifuged at 13,000 rpm for 2 minutes. The supernatant was removed using a pipette and air-dried at room temperature. The resulting pellets were redissolved in 25 μL of 70% aqueous formic acid followed by addition of 25 μL of acetonitrile (100%) and centrifuged at 13,000 rpm for 2 minutes. Following centrifugation, one μL of the supernatant was plated onto a MALDI target plate (Bruker, Singapore). Samples were overlaid with one μL of a-cyano-4-hydroxycinnamic acid (Bruker, Germany)) matrix solution and air dried at room temperature. Samples were analysed for the identification of microbes using MALDI-TOF [18].

To confirm whether any microbes that were isolated from UVC-treated slides were susceptible to UVC treatment, aliquots (100 μl; $10^5$ CFU/mL) were added on probes and on glass slides and dried for 30–45 minutes. These were then treated in the Lumicare disinfectant system with a standard cycle (90 seconds) as described above. Probes and slides were processed to determine numbers of viable microbes as described above.

### *In vitro* testing

To ensure that ultrasound probes used in the current study were uncontaminated at the beginning and after every disinfecting cycle, before each inoculation of *Bacillus* spores, probes were treated in 70% ethanol followed by drying for 20 to 30 minutes. After drying, each probe was suspended in PBS within a sterile test tube and mixed by vortex for 2 minutes at ambient temperature (21˚C). Aliquots of 100 µL were plated onto nutrient agar plates (Oxoid, Basingstoke, UK) and incubated for 24 h at 37˚C. After incubation, the agar plates were examined for bacterial growth.

To confirm the sterility of each ultrasound probe after each disinfectant cycle (UVC treatment), a similar procedure was followed except that *Bacillus* spores (see below; $1–2 \times 10^6$ CFU/mL) were dried on one side of each probe for 45 minutes in the biosafety cabinet before the UVC treatment. After UVC treatment, each ultrasound probe was suspended in PBS within a sterile test tube and mixed by vortex for 2 minutes at ambient temperature (21˚C). Aliquots of 100 µL were plated onto nutrient agar plates (Oxoid) and incubated for 24 h at 37˚C. After incubation, the agar plates were examined for bacterial growth.

Disinfection systems are required to meet the success criteria of ASTM E1837-96 (2007) "Standard Test Method to Determine Efficacy of Disinfection Processes for Reusable Medical Devices (Simulate Use Test)". This test method determines whether medical devices can be disinfected using the test system. The testing includes all steps in processing, such as cleaning and disinfection, and uses multiple inoculations of the same device with a microbe.

Three different ultrasound probes, shown in the Fig 2, were used. These ultrasound probes contain no channels and there is no access to their interiors. The probes were disinfected with 70% ethanol and allowed to fully dry before use.

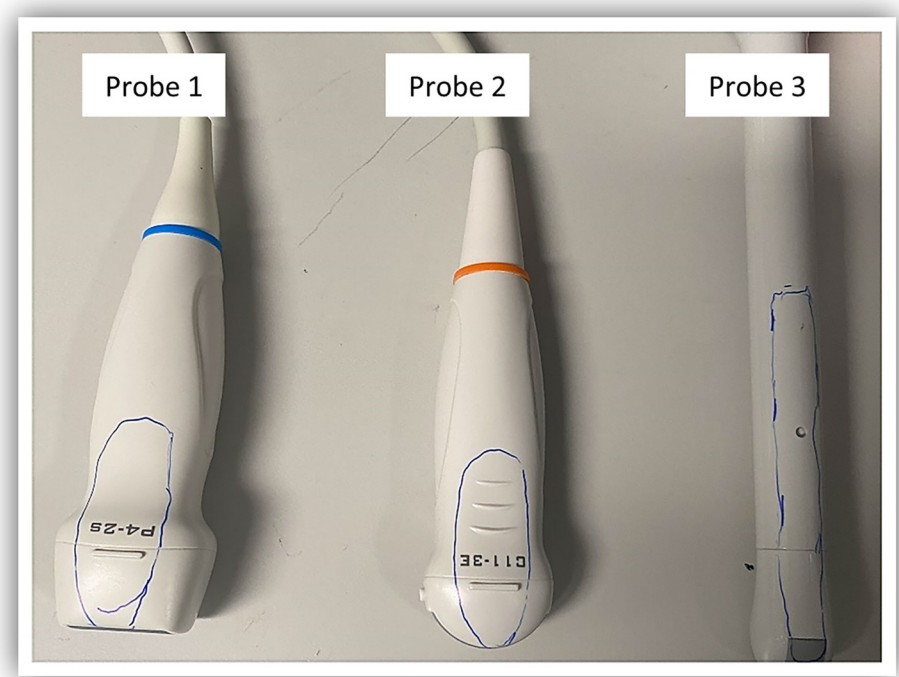

**Fig 2. Ultrasound probes used in the current study.** The blue outlined areas are where the surfaces were either swabbed after use or inoculated with the *Bacillus* spores in the *in vitro* studies.

Vegetative cells of *Bacillus subtilis* ATCC 19659 were cultured in tryptic soya broth (TSB Becton Dickinson and Company, Sydney, NSW, Australia) at 37°C for 16 hours and then transferred to nutrient agar (Oxoid, Basingstoke, UK) supplemented with $MnSO_4.H_2O$ (500 ppm, Sigma-Aldrich, MO, USA, purity $\geq$ 99%) and grown at 37°C for 14 days to allow the generation of spores. Spore formation was confirmed by staining spores with malachite green and microscopic examination. Spores were centrifuged at 5000 rpm and washed in sterile milli Q water and finally resuspended in sterile milli Q water to an OD660 nm of 0.20 yielding 5–9 x $10^8$ spores/mL. To ensure that the spores remained viable, an aliquot, 1.0 mL of a 1x$10^{-7}$ dilution of *B. subtilis* spores was added to a tube containing sterile recovery broth recovery broth (tryptone sodium chloride solution: tryptone pancreatic digest of casein 1.0 g, sodium chloride 8.5 g, water 1000 mL). The tube was incubated at 37° $\pm$ 1°C for up to 24 hours and examined for growth. This bacterium was chosen as members of the genera were isolated from the probes after use, the strain used is recommended for use in several international standards (EN12353, EN14561, AOAC Official Method 966.04) and spores are notoriously resistant to disinfection [19, 20].

To confirm the consistency of the number of *Bacillus subtilis* ATCC 19659 spores, the spores were dissolved in sterile milli Q water or recovery broth (tryptone sodium chloride solution) to an $OD_{660\ nm}$ 0.28, to yield 1.5–5 x $10^9$ spores/mL. Then the spores were further diluted to $10^{-7}$ and $10^{-8}$ dilutions in water or broth. Thereafter, one mL of each dilution was inoculated and spread on nutrient agar plates in duplicate. The plates were incubated at 37°C for up to 24 h. The number of vegetative cells produced from spores were enumerated.

An aliquot, 100 μL, of *B. subtilis* ATCC 19659 spores (5–9 x $10^8$ /mL) were dried onto the surface of the ultrasound probes for 30–45 minutes. After drying, the ultrasound probes were either left at ambient temperature (21°C) for 90 seconds or exposed and treated with UVC for 90 seconds in the Lumicare ONE UVC LED high level disinfection device (Fig 1). Each probe was exposed to the bacterial spores and treated five times sequentially on the same side, with the probes being disinfected with 70% ethanol and dried between runs. The other side of the probe was left uninoculated.

After disinfection, the spores were removed from the ultrasound probes by rubbing vigorously with a sterile swab over the entire inoculated surface followed by suspending the swab in PBS within a sterile test container containing small glass beads (0.5 mm diameter; Sigma Aldrich) within a sterile test container and mixed by vortex for 5–7 minutes at ambient temperature (21°C). The released spores were then diluted in PBS, and aliquots of the diluents plated on the nutrient agar plates (Oxoid) and incubated for 24 h at 37°C. A similar procedure was used for the uninoculated side to confirm sterility. Note, there is no requirement for a neutraliser to be added to media as UVC treatment does not leave residual active ingredients that need to be neutralised during microbial growth.

To ensure the consistency of the number of spores recovered from the probes, $1–2 \times 10^6$ spores/mL of *Bacillus subtilis* suspended sterile milli Q water or diluents were dried on surface of the ultrasound probes for 45 minutes in the biosafety cabinet. The spores were removed from the ultrasound probes by adding small glass beads (0.5 mm diameter; Sigma Aldrich) suspended in PBS within a sterile test tube and mixed by vortex for 2 minutes at ambient temperature (21°C). The released spores were then diluted in PBS, and aliquots of the diluents plated on to nutrient agar plates and incubated for up to 24 h at 37°C. Number of viable vegetative cells produced from spores were enumerated. This process was repeated three times.

To validate the number of spores, a validation suspension was made by diluting the spores in sterile milli Q water or diluents to an OD660 nm of 0.28 and 0.3 to obtain 1.5 x $10^9$ spores/ mL to 5 x $10^9$ spores/mL, respectively. Then spores were diluted in the perspective diluents to obtain between 3.0–1.6 x $10^2$ spores/mL. Each suspension was further diluted to $10^{-1}$ dilution.

Then one mL of each dilution was inoculated and spread onto nutrient agar plates in duplicate. The plates were incubated at 37˚C for 24–48 h. The number vegetative cells produced from spores were enumerated.

To determine the effect of different conditions on the number of spores in the validation suspension, the validation suspension was exposed to selected substances to test for their interference on the final number of spores that germinated on agar plates. These substances were bovine serum albumin (BSA; Bovovgen Biologicals, Vic, AUS, purity $\geq$ 99%) and washed horse red blood cells (RBCs; Bovovgen Biologicals, Vic, AUS, purity $\geq$ 99%), in two different combinations: Clean Condition with BSA (0.3 g) in 100 mL of diluent (Tryptone 1 g; Basingstoke, UK; sodium chloride 8.5 g, Ajax Finchem, AUS; purity of each $\geq$ 99% in 1000 mL of water) and Dirty Condition with 3 g of BSA (Bovovgen Biologicals, Vic, AUS) + 3 mL of defibrinated horse RBCs (Oxoid, Australia) in 97 mL of diluent. One ml of the potential interfering substance was pipetted separately into two separate tubes followed by the addition of one ml of the validation suspension. Tubes were mixed by vortexing and placed in a water bath for 2 minutes at 37˚C. Thereafter, 8 ml of hard water (magnesium chloride19.48 g; Chem supply, SA, AUS and sodium bicarbonate 35.02 g; Sigma, St, Louis, USA, purity of each $\geq$ 99%, in 1000 mL of water) was added into each test tube and the tubes kept in a water bath for 90 seconds at 37˚C. Thereafter, one mL of suspension was inoculated and spread onto nutrient agar plates in duplicate. The plates were incubated at 37˚C for up to 24 h. The number of vegetative cells produced from spores were enumerated.

## Statistical analysis

The numbers of microbes and type of microbes isolated from the transvaginal ultrasound probes after use or after *in vitro* inoculation and then after a disinfection cycle in the Lumicare ONE UVC LED high level disinfection system was repeated two times and compared using nonparametric analysis (Mann Whitney test) with Graph Pad Prism 8 version 8.0.2.

## Results

### *Ex vivo* testing

Microbial growth of UVC treated and Non-UVC treated probes are shown in Fig 3 (a representative image). Rate of contamination was 50% (15/30) on non-UVC treated ultrasound probes. On other hand only one UVC treated probe (3%) was contaminated. Rate of contamination on untreated control probes was significantly higher than the UVC treated probes ($p < 0.001$).

The following microbes were identified (% occurrence) on the probes: *Bacillus pumilus* (39.1%), *Micrococcus luteus* (13%), *Bacillus safensis* (13%), *Pseudomonas luteola* (8.7%), *Bacillus infantis* (4.3%), *Bacillus oceanisediminis* (4.3%), *Bacillus idriences* (4.3%), *Bacillus licheniformis* (4.3%), *Staphylococcus warneri* (4.3%) and *Staphylococcus cohnii* (4.3%). A single UVC treated probe had *Staphylococcus capitis* (4.3%). No fungal species was detected on control or UVC treated probes.

The numbers of bacteria on control (non-UVC treated) probes ranged from 2.4 $\log_{10}$ to 3.8 $\log_{10}$ colony forming units (CFU). *Bacillus pumilus* was present in the highest amounts, 3.8 $\log_{10}$ CFU, followed by 3.7 $\log_{10}$ CFU of *Bacillus safensis* (Table 1). *Micrococcus luteus* and *Pseudomonas luteola* were present at lower numbers (2.4 $\log_{10}$ CFU; Table 1).

After UVC treatment, *Staphylococcus capitis* was grown from only one probe. This may have been due to contamination of the probe after treatment. To check whether the bacterial strain was susceptible to UVC treatment, a probe and glass slides were contaminated with the strain in the laboratory and then treated with the Lumicare system. Like other bacteria, the

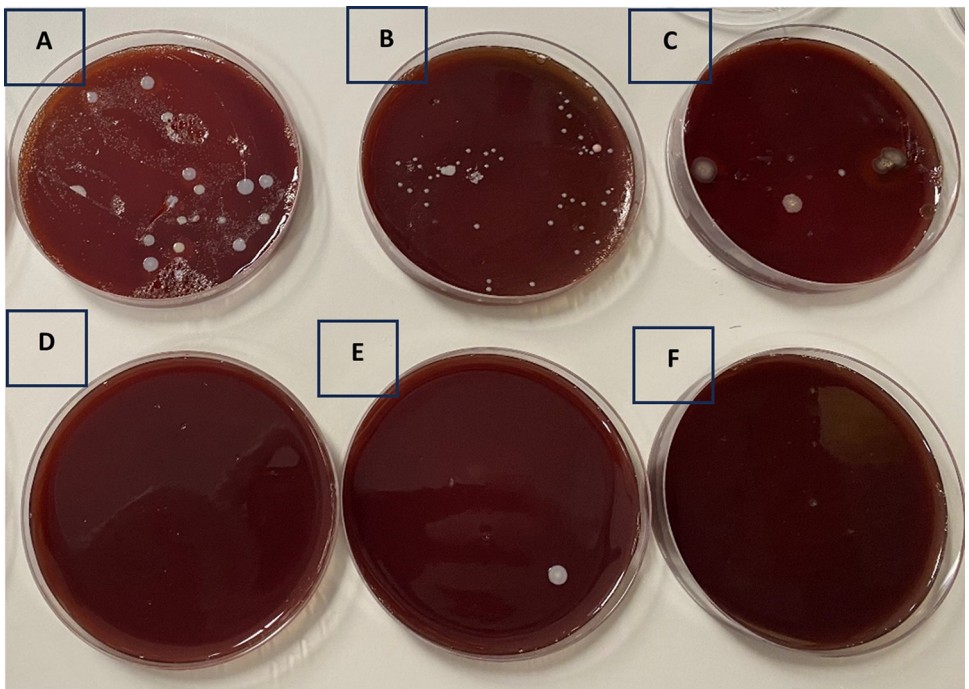

**Fig 3. Detection of bacteria on chocolate blood agar.** Fig 3A-3C represent the bacterial colonies on the plate of non-UVC treated probe while Fig 3D-3F represent the bacterial colonies on the plates of UVC treated probe.

UVC system reduced the numbers of this strain of *S. capitis* on the probes and slides to below the detection limits. The UVC system produced 3.13 $\log_{10}$ killing on the probes and 4.02 $\log_{10}$ on glass slides.

The sheaths which were used to cover the probes were found contaminated. Both the sheaths were contaminated with *Bacillus subtilis*. The numbers of *Bacillus subtilis* on sheaths were 2.8 $\log_{10}$ CFU.

### *In vitro* testing

As the most common bacteria grown from the non-UVC treated probes were *Bacillus* spp., the in vitro tests used a standard strain of this genera.

No spores germinated from the probes after disinfecting with 70% ethanol, showing that this disinfection of the probes was appropriate to decontaminate probes prior to use in the study. Also, there was no visible sign of bacterial growth in the sterile recovery broth after incubation for up to 24 h, demonstrating that the broth remained sterile. However, there was visible growth in the tube that contained 1.0 mL of a $1 \times 10^{-7}$ dilution of spores, demonstrating that the spores were viable. The OD660nm 0.28 and 0.3 of spore suspensions yielded $1.18 \times 10^9$ CFU/mL and $1.39 \times 10^9$ CFU/mL in milli Q water and diluents respectively. After drying, $1–2 \times 10^6$ spores/mL of *Bacillus subtilis* onto the endoscope probes, $3.20 \pm 0.47 \times 10^5$ could be recovered, indicating $\leq 0.83 \log_{10}$ reduction in the number of spores after drying.

The number of cells that germinated from spores in the validation suspension were 53 and 51 in milli Q water and validation suspension respectively. Also, between 60 to 80 cells germinated from the spores in the clean and dirty conditions. These results conformed to the requirements of European standard EN14561 (Chemical disinfectants and antiseptics. Quantitative carrier test for the evaluation of bactericidal activity for instruments used in the medical area).

**Table 1. Numbers of bacteria identified on untreated control and UVC treated probes.**

| Sample Numbers | Untreated control sides | | UVC treated sides | | P value |
|---|---|---|---|---|---|
| | Bacterial type | Numbers of bacteria (CFU) (Log$_{10}$ CFU) | Bacterial type | Numbers of bacteria (CFU) (Log$_{10}$ CFU) | |
| Sample 1 | *Staphylococcus warneri* | 400 (2.6) | - | 0 | p<0.001 |
| Sample 5 | *Micrococcus luteus* | 580 (2.8) | - | 0 | |
| | *Bacillus infantis* | 4300 (3.6) | - | 0 | |
| | *Bacillus pumilus* | 4300 (3.6) | - | 0 | |
| Sample 9 | *Micrococcus luteus* | 2200 (3.3) | - | 0 | |
| | *Bacillus pumilus* | 460 (2.7) | - | 0 | |
| | *Staphylococcus cohnii* | 2100 (3.3) | - | 0 | |
| Sample 14 | *Bacillus oceanisediminis* | 1700 (3.2) | - | 0 | |
| | *Micrococcus luteus* | 240 (2.4) | - | 0 | |
| Sample 15 | *Bacillus pumilus* | 1900 (3.3) | - | 0 | |
| | *Bacillus idriences* | 1540 (3.2) | - | 0 | |
| Sample 16 | *Bacillus pumilus* | 6020 (3.8) | - | 0 | |
| | *Bacillus safensis* | 1800 (3.3) | - | 0 | |
| Sample 17 | *Bacillus pumilus* | 1760 (3.2) | - | 0 | |
| Sample 18 | *Bacillus licheniformis* | 2200 (3.3) | - | 0 | |
| Sample 19 | *Pseudomonas luteola* | 240 (2.4) | - | 0 | |
| | *Bacillus pumilus* | 4000 (3.6) | - | 0 | |
| Sample 20 | *Pseudomonas luteola* | 6000 (3.8) | - | 0 | |
| Sample 21 | *Bacillus pumilus* | 3000 (3.5) | - | 0 | |
| Sample 22 | *Bacillus pumilus* | 6020 (3.8) | - | 0 | |
| Sample 25 | *Bacillus safensis* | 4500 (3.7) | - | 0 | |
| Sample 28 | *Bacillus safensis* | 240 (2.4) | - | 0 | |
| Sample 30 | *Bacillus pumilus* | 4400 (3.6) | *Staphylococcus capitis* | 720 (2.9) | |

The table only includes sequentially listed sample numbers (1–30) that exhibited bacterial growth while those sample numbers were omitted where bacterial growth was absent.

In the absence of UVC treatment, the number of cells that germinated from Probe 1 (Fig 2) was $1.73 \times 10^7 \pm 0.26$, from Probe 2 (Fig 2) was $1.29 \times 10^7 \pm 0.31$ and from Probe 3 (Fig 2) was $1.43 \times 10^7 \pm 0.29$ CFU/mL. After UVC treatment, no viable spores were recovered from any of the probes each time they were tested. Thus, the UVC disinfection system was able to reduce the numbers of *Bacillus subtilis* ATCC 19659 spores on probes after 5 standard disinfection cycles by >7 log$_{10}$ CFU.

## Discussion

The UVC LED system, Lumicare ONE UVC LED high level disinfection system, was highly effective at reducing the microbial load on transvaginal ultrasound probes. This system was able to disinfect the probes and reduce the number of bacteria on 97% of probes (only one probe had microbes–*S. capitis*—cultured after the UVC disinfection cycle) by more than 3.8 log$_{10}$ CFU. This is slightly greater than the reported efficacy of a currently marketed automated system that uses hydrogen peroxide vapour which resulted in 91.4% of probes having no microbial growth after use [14]. For the strain of *S. capitis* isolated from the one probe after UVC disinfection in the current study, in laboratory studies this strain was reduced to below detectable limits (>3.13 log$_{10}$ reduction) on the probe and glass slides. This most likely indicates that the *S. capitis* was a contaminant introduced after UVC treatment probably due to

mishandling of the probe. *S. capitis* was not cultured from probes after use before UVC treatment, suggesting its absence in the vagina [21].

UVC rapidly inactivates microbial cells by several mechanisms including inducing synthesis of reactive oxygen species and directly damaging DNA [22, 23] and unlike for traditional antibiotics, microbes appear to have a low ability to become resistance to UVC [22] UVC has been shown to be an effective system to supplement disinfection of in dentistry [24, 25] and UVC has been shown to be effective in improving food safety [26]. Furthermore, UVC delivered by LEDs has been shown to provide the highest antimicrobial activity compared to UVA or UVB [23].

Like present study, UVC has been used previously to decontaminate transvaginal probes [7, 27]. Consistent with another study [17], the present study did not find any fungal species associated with the probes. The predominance of bacteria of the *Bacillus* genus probably indicates that the transvaginal probes were not contaminated during use, as these bacteria are not part of the normal vaginal microbiota which is usually composed of members of the *Lactobacillus* genus as well as genera *Gardnerella*, *Prevotella*, *Atopobium*, *Sneathia*, *Megasphaera* and *Peptoniphilus* [28]. A previous study, in which *Acinetobacter* spp has been found associated with the probes, also suggested that this bacteria was introduced from external sources as this is not part of normal vaginal microbiota [29]. Contamination of probes can occur via the skin microbiota. Common skin microbes such as coagulase-negative staphylococci (which includes *S. capitis*, *S. warneri* and *S. cohnii*) and *Micrococcus* sp. or those found in the environment (e.g. *Pseudomonas* spp.) have also previously been found on transvaginal probes [7]. The findings from the current study, and those of previous studies as outlined above, indicate that contamination can occur from several sources. Therefore, it is essential to take necessary precautions, including ensuring that the person collecting the samples has been adequately trained.

*Bacillus* sp. are more commonly associated with air, soil or water. This suggests that the ultrasound probes were contaminated outside of the body. However, they may be associated with the human gut [30], which might also suggest some faecal contamination during use. Contamination of sheaths covering probes with *Bacillus subtilis* indicated that the probes were most likely contaminated with bacteria from the sheaths that cover the probes before use. The difference in the species of *Bacillus* identified within the studies may be due to the use of different batches of non-latex sheaths, or that the MALDI identification system poorly speciates members of the *Bacillus* genera. Given that the probes can be contaminated using non-sterile sheaths, we recommend that users consider the use of sterilised probes to reduce any risk introducing microbes into the vagina during use. It has been recommended to use sterile single use covers [5], and current data reinforces this as the study demonstrated that when non-sterile covers were used there was significant contamination of probes. However, if the probes are disinfected, as in the current protocol, no viable microbes could be cultured from almost all probes, indicating the need for sterilisation and the effectiveness of the current UVC system.

The current study demonstrated that the ultrasound probes could repeatedly meet the standard for high level disinfection (ASTM E1837-96) by reducing $> 10^6 \log_{10}$ *B. subtilis* spores after repeated inoculation of their surfaces.

The current study used UV-C delivered via LED lights with wavelengths ranging from 265nm to 275nm. This contrasts with UV-C delivered using mercury tubes which emit UV-C radiation at a wavelength of 254nm. This might be important as the optimum wavelength for inactivating *Escherichia coli* has been shown to be 265nm, which was approximately 15% more effective than UV-C of 254nm [31]. The optimum wavelength for destroying *Cryptosporidium parvum* oocysts is 271nm and this was approximately 15% more effective than 254nm [32]. The reason for the superior performance of wavelengths of between 265 and 270nm probably due to thymine and cytosine having strong absorbance peaks near 265nm in the UV

absorption spectra [31]. Another advantage of the current LED system was its low temperature inside the device whilst it was running. Owing to the inherent characteristics of mercury UV-C tubes, a substantial build-up of heat within the disinfection chamber occurs during the disinfection cycle, which can result in the ultrasound transducer being heated. This elevated temperature presents a potential concern, as it may pose a thermal risk which may affect the operational lifespan of ultrasound transducers. Furthermore, the UN Minamata Convention on Mercury (http://www.mercuryconvention.org) sets down controlling measures over a variety of products containing mercury, and for some the manufacture, import and export will be prohibited.

Limitations of this study include evaluating probes from only one clinical centre as the spectrum of contaminating microbes may be different between centres. Another limitation is that the study did not do a direct comparison of the UVC system with another commercially available system, which would have allowed a direct comparison of the efficacy of the systems. Additionally, the study did not evaluate the ability of the system to reduce the numbers of viruses that may be present on probes. This will be investigated in future studies, as it is possible that transvaginal probes serve as a vehicle for transmission of human papillomavirus [5] which can then cause cervical cancer.

## Conclusions

In conclusion, this study has demonstrated that UVC light supplied by LED lights can be used to produce high level disinfection of ultrasonic probes used for TVUS. This system provides rapid high-level disinfection of transvaginal probes.

## Author Contributions

**Conceptualization:** Mark D. P. Willcox.

**Formal analysis:** Muhammad Yasir, Mark D. P. Willcox.

**Funding acquisition:** Mark D. P. Willcox.

**Methodology:** Muhammad Yasir, Mark D. P. Willcox.

**Project administration:** Mark D. P. Willcox.

**Resources:** Mark D. P. Willcox.

**Supervision:** Mark D. P. Willcox.

**Writing – original draft:** Muhammad Yasir, Mark D. P. Willcox.

**Writing – review & editing:** Muhammad Yasir, Mark D. P. Willcox.

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
