## [Decision Letter · Decision Letter 0]

24 Nov 2023

PONE-D-23-30694The use of ultraviolet light generated from light-emitting diodes for the disinfection of transvaginal ultrasound probesPLOS ONE

Dear Dr. Yasir,

Thank you for submitting your manuscript to PLOS ONE. After careful consideration, we feel that it has merit but does not fully meet PLOS ONE’s publication criteria as it currently stands. Therefore, we invite you to submit a revised version of the manuscript that addresses the points raised during the review process.

We look forward to receiving your revised manuscript.

Kind regards,

Amitava Mukherjee, ME, Ph.D.

Academic Editor

PLOS ONE

“There is no competing interest. Lumicare One holds the right of this device.”

6. Thank you for stating the following in the Funding Section of your manuscript:

“This research was funded by Lumicare, Sydney, Australia.”

“The author(s) received no specific funding for this work”

7. Your ethics statement should only appear in the Methods section of your manuscript. If your ethics statement is written in any section besides the Methods, please delete it from any other section.

Reviewers' comments:

Reviewer's Responses to Questions

**Comments to the Author**

1. Is the manuscript technically sound, and do the data support the conclusions?

Reviewer #1: No

2. Has the statistical analysis been performed appropriately and rigorously? 

Reviewer #1: I Don't Know

3. Have the authors made all data underlying the findings in their manuscript fully available?

Reviewer #1: Yes

4. Is the manuscript presented in an intelligible fashion and written in standard English?

Reviewer #1: Yes

5. Review Comments to the Author

Reviewer #1: The results of the studies “The use of ultraviolet light generated from light-emitting diodes for the disinfection of transvaginal ultrasound probes” are predictable. UVC radiation is effective in removing microorganisms. The presented conclusions are of practical importance, therefore I propose to improve some aspects of the manuscript.

1. Line 187, 244, 298, 301, 302 What is the purity of reagents? Where do the reagents come from? The reagents must be described in sufficient detail.

2. How many repetitions of the experimental measurements were performed?

3. Give the name and version of any software used, e.g. in Statistical Analysis

4. I recommend adding Bacteria Detection Images to the manuscript. This visualization will complement the study.

5. Some samples are missing from the experiment description, e.g. Sample 2, 3, 4, .... What does this mean? Are these reference tests?

6. PLOS authors have the option to publish the peer review history of their article (what does this mean?). If published, this will include your full peer review and any attached files.

Reviewer #1: No

---

## [Author Response · Author response to Decision Letter 0]

1 Jan 2024

Comment 1. Please ensure that your manuscript meets PLOS ONE's style requirements, including those for file naming. The PLOS ONE style templates can be found at

https://journals.plos.org/plosone/s/file?id=wjVg/PLOSOne_formatting_sample_main_body.pdfandttps://journals.plos.org/plosone/s/file?id=ba62/PLOSOne_formatting_sample_title_authors_affiliations.pdf.

Reply: The authors have carefully formatted the manuscript according to PLOS One guidelines. 

Comments 2. Please provide additional details regarding participant consent. In the ethics statement in the Methods and online submission information, please ensure that you have specified what type you obtained (for instance, written or verbal, and if verbal, how it was documented and witnessed). If your study included minors, state whether you obtained consent from parents or guardians. If the need for consent was waived by the ethics committee, please include this information.

Reply: Patients consent was obtained on consent forms that included their names and signatures. We have written this statement in method section in detail. 

Comment 3. We suggest you thoroughly copyedit your manuscript for language usage, spelling, and grammar. If you do not know anyone who can help you do this, you may wish to consider employing a professional scientific editing service.

Whilst you may use any professional scientific editing service of your choice, PLOS has partnered with both American Journal Experts (AJE) and Editage to provide discounted services to PLOS authors. Both organizations have experience helping authors meet PLOS guidelines and can provide language editing, translation, manuscript formatting, and figure formatting to ensure your manuscript meets our submission guidelines. Upon resubmission, please provide the following: The name of the colleague or the details of the professional service that edited your manuscript A copy of your manuscript showing your changes by either highlighting them or using track changes (uploaded as a *supporting information* file). A clean copy of the edited manuscript (uploaded as the new *manuscript* file).

Reply: Professor Mark D.P. Willcox edited the revised manuscript for English. All the changes in the revised manuscript are highlighted red with track changes the revised manuscript and another clean copy of manuscript is also provided.

Comment 4. Thank you for stating the following in your Competing Interests section: 

“There is no competing interest. Lumicare One holds the right of this device.” Please complete your Competing Interests on the online submission form to state any Competing Interests. If you have no competing interests, please state "The authors have declared that no competing interests exist.", as detailed online in our guide for authors at http://journals.plos.org/plosone/s/submit-now. This information should be included in your cover letter; we will change the online submission form on your behalf.

Reply: The authors have declared that no competing interests exist. Please change this in the online submission form on the authors behalf. 

Comment 5. We note that you have stated that you will provide repository information for your data at acceptance. Should your manuscript be accepted for publication, we will hold it until you provide the relevant accession numbers or DOIs necessary to access your data. If you wish to make changes to your Data Availability statement, please describe these changes in your cover letter and we will update your Data Availability statement to reflect the information you provide.

Reply: Sorry, this is our mistake. There is no repository information that need to be supplied. I have supplied all the information with my manuscript. Please correct this on the authors behalf. 

Comment 6. We note that you have provided funding information that is not currently declared in your Funding Statement. However, funding information should not appear in the Acknowledgments section or other areas of your manuscript. We will only publish funding information present in the Funding Statement section of the online submission form.

“The author(s) received no specific funding for this work”. Please include your amended statements within your cover letter; we will change the online submission form on your behalf.

Reply: Sorry for inconvenience. Yes, this is correct that “This research was funded by Lumicare, Sydney, Australia”. Please update this information online submission form.

Comment 7. Your ethics statement should only appear in the Methods section of your manuscript. If your ethics statement is written in any section besides the Methods, please delete it from any other section. 

Reply: We have written ethics statement in the method section and deleted from elsewhere in the manuscript. 

 Response to Reviewer 

Reviewer 1:

Reviewer #1: The results of the studies “The use of ultraviolet light generated from light-emitting diodes for the disinfection of transvaginal ultrasound probes” are predictable. UVC radiation is effective in removing microorganisms. The presented conclusions are of practical importance; therefore I propose to improve some aspects of the manuscript.

Reply: Thank you for appreciation our work. We have addressed the reviewer comment comments and added the information’s in the revised manuscript. 

Comments: 1. Line 187, 244, 298, 301, 302 What is the purity of reagents? Where do the reagents come from? The reagents must be described in sufficient detail.

Reply: 1. We have provided purity (where applicable) and the source of each reagent in detail, please see in the revised manuscript:

1. Page # 7, Line # 187-189: autoclaved sterile phosphate buffered saline (PBS; 8 g NaCl, Ajax Finchem, AUS; 0.2 g KCl, Ajax Finchem, AUS; 1.4 g Na2HPO4, Chem Supply, AUS; 0.24 g KH2PO4, Chem Supply, AUS; purity of each ≥ 99 %, in 1000 mL of milli Q water).

2. Page # 10, Line # 245-246: (Oxoid, Basingstoke, UK) supplemented with MnSO4.H2O (500 ppm, Sigma-Aldrich, MO, USA, purity ≥ 99 %).

3. Page # 12, Line # 298-303: (BSA; Bovovgen Biologicals, Vic, AUS, purity ≥ 99%) and washed horse red blood cells (RBCs; Bovovgen Biologicals, Vic, AUS, purity ≥ 99% ), in two different combinations: Clean Condition with BSA (0.3 g ) in 100 mL of diluent (Tryptone 1 g; Basingstoke, UK; sodium chloride 8.5 g, Ajax Finchem, AUS; purity of each ≥ 99% in 1000 mL of water) and Dirty Condition with 3 g of BSA (Bovovgen Biologicals, Vic, AUS) 

4. Page # 12, Line # 298-303: (magnesium chloride19.48 g; Chem supply, SA, AUS and sodium bicarbonate 35.02 g; Sigma, St, Louis, USA, purity of each ≥ 99 %, in 1000 mL of water)

Comments: 2. How many repetitions of the experimental measurements were performed?

Reply: Each experiment was in performed in duplicate. Please see in the revised manuscript Page # 13, Line # 315. 

Comments: 3. Give the name and version of any software used, e.g. in Statistical Analysis

Reply: Graph pad prism 8 version 8.0.2 was used for statical analysis and written in revised manuscript (Page # 13, line# 307-308). 

Comments: 4. I recommend adding Bacteria Detection Images to the manuscript. This visualization will complement the study.

Reply: Thank you for your suggestions. We have included following figure in the manuscript. (Page # 14).

Figure 3. Detection of bacteria on chocolate blood agar. Figure 3A-C represent the bacterial colonies on the plate of non-UVC treated probe while Figure 3D-F represent the bacterial colonies on the plates of UVC treated probe. 

Comments: 5. Some samples are missing from the experiment description, e.g. Sample 2, 3, 4, .... What does this mean? Are these reference tests?

Reply: We apologies for inconvenience. We have clarified this in the revised manuscript by adding the information’s in the Table legend. 

Please see Page # 13, line# 307-308. “The table only includes sequentially listed sample numbers (1-30) that exhibited bacterial growth while those sample numbers were omitted where bacterial growth was absent”.

---

## [Decision Letter · Decision Letter 1]

25 Jan 2024

The use of ultraviolet light generated from light-emitting diodes for the disinfection of transvaginal ultrasound probes

PONE-D-23-30694R1

Dear Dr. Yasir,

We’re pleased to inform you that your manuscript has been judged scientifically suitable for publication and will be formally accepted for publication once it meets all outstanding technical requirements.

Kind regards,

Amitava Mukherjee, ME, Ph.D.

Academic Editor

PLOS ONE

Additional Editor Comments (optional):

Reviewers' comments:

Reviewer's Responses to Questions

**Comments to the Author**

1. If the authors have adequately addressed your comments raised in a previous round of review and you feel that this manuscript is now acceptable for publication, you may indicate that here to bypass the “Comments to the Author” section, enter your conflict of interest statement in the “Confidential to Editor” section, and submit your "Accept" recommendation.

Reviewer #1: All comments have been addressed

2. Is the manuscript technically sound, and do the data support the conclusions?

Reviewer #1: Yes

3. Has the statistical analysis been performed appropriately and rigorously? 

Reviewer #1: Yes

4. Have the authors made all data underlying the findings in their manuscript fully available?

Reviewer #1: Yes

5. Is the manuscript presented in an intelligible fashion and written in standard English?

Reviewer #1: Yes

6. Review Comments to the Author

Reviewer #1: The manuscript has been much strengthened by the additional data. All the concerns of mine are solved in the revised manuscript.

7. PLOS authors have the option to publish the peer review history of their article (what does this mean?). If published, this will include your full peer review and any attached files.

Reviewer #1: No

---

## [Editor Report · Acceptance letter]

14 Feb 2024

PONE-D-23-30694R1 

PLOS ONE

Dear Dr. Yasir, 

I'm pleased to inform you that your manuscript has been deemed suitable for publication in PLOS ONE. Congratulations! Your manuscript is now being handed over to our production team.

Kind regards, 

on behalf of

Professor Dr. Amitava Mukherjee 

Academic Editor

PLOS ONE